# Initial Experience of the Levodopa–Entacapone–Carbidopa Intestinal Gel in Clinical Practice

**DOI:** 10.3390/jpm11040254

**Published:** 2021-03-31

**Authors:** Mezin Öthman, Erik Widman, Ingela Nygren, Dag Nyholm

**Affiliations:** Department of Neuroscience, Neurology, Uppsala University, SE-75185 Uppsala, Sweden; mezin.othman@akademiska.se (M.Ö.); erikwidman1@hotmail.com (E.W.); ingela.nygren@neuro.uu.se (I.N.)

**Keywords:** levodopa infusion therapy, carbidopa, entacapone, Parkinson’s disease, pump

## Abstract

Patients in fluctuating stages of Parkinson’s disease (PD) require device-aided treatments. Continuous infusion of levodopa–carbidopa intestinal gel (LCIG) is a well-proven option in clinical practice. We now report the first clinical experience of levodopa–entacapone–carbidopa intestinal gel (LECIG) therapy. An observational study of the first patients to start LECIG in our clinic was performed. Twenty-four patients (11 females, 13 males) were included. The median age was 71.5 years, and the median duration since PD diagnosis was 15.5 years. The median treatment duration was 305 days. Median doses were: 6.0 mL as morning dose, 2.5 mL/h as infusion rate, and 1.0 mL as extra dose. Half of the patients were switched directly from LCIG. These patients express improvements in the size and weight of the pump. Furthermore, most of them considered the new pump to be improved regarding user-friendliness. Six patients discontinued LECIG, three due to diarrhea, one due to hallucinations and two deceased (one cardiac arrest and one COVID-19). LECIG has shown to be possible to use in patients with PD, efficacy and safety as expected. Patients are generally happy with the size and usability of the pump, but some technical improvements of the software are warranted, as well as larger, prospective studies.

## 1. Introduction

Continuous infusion of levodopa–carbidopa intestinal gel (LCIG) is approved in many countries for the treatment of motor fluctuations in Parkinson’s disease (PD) [1]. The infusion via an intestinal tube immediately provides levodopa at the absorption site, rendering stable levodopa concentrations in plasma and thereby continuous drug delivery to the brain [2]. LCIG efficiently reduces motor fluctuations and dyskinesias [3], but also some non-motor fluctuations [4]. It thereby increases the quality of life in patients with fluctuating responses to oral dopaminergic drugs [5] and has the potential to serve the patients for many years, like the other device-aided therapies, subcutaneous apomorphine infusion and deep brain stimulation (DBS) [6].

However, there are some limitations and concerns with LCIG therapy. For example, many patients complain about the size and weight of the pump, which has not been changed since the method was introduced in Sweden in the 1990s, although two different models of the pump have been used [7,8]. Another potential complication during LCIG therapy is the development of peripheral neuropathy. The mechanisms are not fully understood, but levodopa is involved in the B-vitamins metabolization, and high doses of levodopa with low levels of B-vitamins and/or high levels of homocysteine are associated with a higher risk for polyneuropathy in PD patients [9]. Inhibitors of catechol-O-methyltransferase (COMT) have been used as an oral therapy adjunctive to levodopa since the early 2000s. COMT inhibitors decrease the peripheral metabolization of levodopa and allow higher amounts of levodopa to reach the brain for conversion to dopamine [10]. Therefore, combining oral levodopa with a COMT inhibitor may potentiate levodopa’s effect by treating wearing-off fluctuations, but this strategy may also cause dyskinesias [11]. Interestingly, COMT inhibitors also prevent the methionine cycle from using B-vitamins to form homocysteine [12].

The idea to combine LCIG and the oral COMT inhibitors entacapone and tolcapone was tested in a previous study, where the dose of LCIG could be reduced by 20% with preserved stability of levodopa concentrations and motor function [13]. This led to the development of a levodopa–entacapone–carbidopa intestinal gel (LECIG). Entacapone was chosen over tolcapone due to its safety profile, without hepatotoxicity. LECIG was compared with LCIG in a randomized crossover trial of 1 + 1 day, showing similar systemic exposure and stable motor outcome [14]. LECIG was approved by the Swedish Medical Products Agency in 2018 and reimbursed in Sweden from 2019, but no studies of LECIG use in clinical practice have been published so far. We now report our initial experience, based on the clinical use of LECIG and patient-reported outcome. The objectives were to assess patient-reported efficacy, tolerability and usability of LECIG during the first year of therapy. The hypotheses were that long-term LECIG infusion was possible and that patients switching from LCIG to LECIG would be content with the size of the new pump.

## 2. Methods

This is an observational clinical study of patients started on LECIG therapy at Uppsala University Hospital, Sweden, following the introduction on the market, i.e., from June 2019, until January 2021. The study was approved by the Swedish Ethical Review Authority.

The LECIG formulation contains levodopa (20 mg/mL), entacapone (20 mg/mL) and carbidopa (5 mg/mL) in a methylcellulose gel (Lecigon^®^, LobSor Pharmaceuticals AB, Uppsala, Sweden). The gel is contained in a syringe of 47 mL, which is connected to a portable pump (Canè, Rivoli, Italy), together measuring 55 × 150 mm, with a total weight of 227 g (Figure 1). The pump delivers LECIG to the proximal small intestine via an intestinal tube, usually as a jejunal extension tube within a percutaneous endoscopic gastrostomy (PEG-J) or via a radiologically placed transcutaneous port (T-port) [15]. The same PEG-J tubing system as with LCIG can be used, but an ENFit^®^ connector (The Global Enteral Device Supplier Association (GEDSA), Powell, OH, USA) is required between the syringe and the PEG-J.

All patients with idiopathic PD treated with LECIG at Uppsala University Hospital were screened for eligibility (*n* = 24). These were contacted by letter. The letter included an information sheet, an informed consent form and patient questionnaires. A few patients were informed and included during a regular visit to the clinic. Patients who did not respond to the letter were contacted by telephone.

All patients were followed up according to the usual clinical routine, which means a visit, either as outpatient or inpatient, every 6 months, following Swedish national guidelines for the care of PD [16]. For de novo patients, contact with a PD nurse within one month after LECIG initiation and a first hospital follow-up visit 2 months later is routine.

The patient questionnaires sought information about the experienced change in symptoms, user-friendliness of the drug delivery system, patient-reported activities of daily living, patient-reported health-related quality of life and, for patients switched from LCIG to LECIG, questions comparing these two therapies.

The medical records of the patients were examined regarding gender, age, duration since PD diagnosis, treatment duration with LECIG, concomitant PD medications and dosage before and during LECIG infusion therapy, complications, technical problems and coexisting treatment with deep brain stimulation (DBS). The levodopa dose given in the form of LECIG was recorded at the first and latest/last treatment days. The calculation of levodopa dosage in LCIG and LECIG, when the duration of infusion therapy was not specified in the medical records, used a 15 h duration of daytime treatment [17]. Levodopa equivalent dose (LED) was calculated by using conversion factors described by Tomlinson and colleagues in 2010 [18]. LED for LECIG was calculated by using the conversion factor obtained by Senek and colleagues in 2017 [19].

The utilization of multiple flows of LECIG therapy was documented. Blood tests for vitamin B12, folic acid and homocysteine were registered. The number of hospitalizations, visits to the neurology clinic and the number of messages or telephone contacts with the clinic were documented. Furthermore, the titration duration and the use of PEG-J or T-port for intestinal access were examined as well. Descriptive data were reported, and statistical analyses were considered irrelevant in this small, open, observational report.

## 3. Results

Written consents and survey answers were obtained from 21 out of 24 eligible patients. Three patients had died prior to the study (two of them under treatment with LECIG and one after discontinuing LECIG therapy), thus giving no written consent or survey answers, but data from medical records were used. Thus, 100% of patients exposed to LECIG were included in this report.

The demographics of the study participants are shown in Table 1. Two patients were treated with DBS of the subthalamic nuclei (STN) combined with LECIG, and one had previously removed the DBS system due to severe infection. Twelve patients (50%) switched directly from LCIG. Three more patients were previously treated with LCIG. Two patients were switched from apomorphine infusion therapy to LECIG. Three patients were switched from levodopa/carbidopa micro tablets (LC-5) [19] to LECIG.

The characteristics of LECIG infusion therapy are summarized in Table 2. The initiation and titration of LECIG therapy were done at the ward for 17 patients over a median of 3 days. Patients without existing PEG-J used a nasointestinal tube. The patients were then scheduled for PEG-J or T-port tube insertion. One patient discontinued the treatment before PEG-J tube insertion due to hallucinations. In six patients with prior LCIG treatment, the switching of treatment to LECIG was done as outpatients over the day. One patient with prior LCIG treatment did the titration at home with a support nurse in place and physician via video link. Twenty-one patients used a PEG-J system, while two used a T-Port for intestinal access. One patient experienced complications related to the pump (breakage), while four patients experienced complications related to the intestinal access (stoma infection, dislocation of intestinal tube, tear in PEG tube, leakage and occlusion of the intestinal tube). These complications were resolved and did not lead to the discontinuation of treatment. Three other patients discontinued the LECIG treatment because of diarrhea. In addition to these, three patients experienced side effects not leading to discontinuation (sweating, nausea, freezing and hallucinations). Three patients died (one due to COVID-19, one due to pulmonary embolism and cardiac arrest, and one after discontinuing LECIG therapy). Five patients were admitted to the hospital during their LECIG treatment. The median number of visits to the neurological clinic as outpatients during the 10-month median treatment time (range 0.5–19 months) was 0.5 (0–4), telephone consultations were 11 (0–92), and messages to the outpatient clinic via 1177.se health service were 2 (0–10). Two patients used two different flow rates during the day (slightly lower in the afternoon), and four patients used one flow rate during daytime and one at nighttime. However, one patient discontinued the night flow after a couple of weeks.

The numbers of patients treated with other dopaminergic PD medications prior to and during LECIG are shown in Table 3.

Table 4 shows the median flow rates during LECIG treatment at initiation (first) and the latest flow rates (latest). Four patients used continuous flow night time.

Table 5 shows the median doses of LCIG in patients previously treated with LCIG, doses of LECIG at initiation (first) and at the latest follow-up. At initiation, a median of 100% of the morning dose was used, while the continuous infusion rate was 76% of the LCIG dose. Comparing the latest/last dosage of LECIG with prior LCIG dosage, the morning bolus and daytime flow rate were 92% and 68%.

The non-dopaminergic PD treatments prior to and during LECIG treatment are presented in Table 6.

The median daily dose of L-DOPA prior to LECIG was 1210 mg (range, 435–2400 mg) and fell to 1040 mg (range, 370–2000 mg) at the initiation of LECIG and 1080 mg (range, 510–1822 mg) at the end of the study. Median LEDD was 1523 mg (range, 535–2810 mg) prior to LECIG treatment and 1365 mg (range, 798–2478 mg) at the latest dosage.

In the questionnaire, most of the patients who had not used levodopa infusion before (*n* = 10) perceived that the symptoms had improved (70%), see Figure 2. The most common perception of the effect of LECIG on PD symptoms from those who switched from LCIG (*n* = 11) was that there was no change (45%).

Most patients reported improvement in the ability to perform daily activities and in their self-rated quality of life (Table 7).

Most patients thought that it worked well to handle the different aspects of the pump. However, 13 patients answered that they did not always have time to press the buttons before the menu switched back, and 2 other patients experienced this as a big problem.

A majority of the patients previously treated with LCIG regarded the new pump to be improved both with respect to user-friendliness and to changing cassette/syringe. All the patients thought that the size was improved (Figure 3).

## 4. Discussion

This is the first report on the clinical use of LECIG. It shows that LECIG therapy is generally well tolerated and possible to use as a long-term therapy. Furthermore, it has good patient-reported efficacy. The new pump shows good user-friendliness, and patients switching from LCIG to LECIG are content with the size of the pump.

Patients on LCIG are usually happy with their treatment and tend to stay on the treatment for several years [20,21]. The most common complaint, in our experience, is the size and the weight of the pump, and patients usually compare the pump with first-generation mobile phones, an area where product development has been immense during the period from the introduction of LCIG. The LCIG pump (Smiths Medical, Minneapolis, MN, USA) and cassette (100 mL, AbbVie Ltd., Chicago, IL, USA) measure 100 × 197 mm, and the weight of the LCIG pump system with a full cassette is approximately 550 g, which is more than twice the weight of the LECIG pump, including full syringe. The new pump was clearly a major reason for patients in the present study to choose LECIG before LCIG. It is a strength of the present study that all patients who were exposed to LECIG during the first year of prescription consented to participate, with the exception of the deceased patients.

One patient, who unexpectedly died from pulmonary embolism and cardiac arrest after 47 days of LECIG infusion, had previously been admitted to hospital for several months due to severe postoperative complications of DBS surgery, including explantation of electrodes and device, and had been treated with apomorphine infusion until switching to LECIG. Apart from this case, no unexpected adverse events arose. Diarrhea is a well-known side-effect of entacapone [22]. We decided to let patients with a history of possible entacapone-related diarrhea try LECIG to see if the mode of administration mattered. It now seems that entacapone causes diarrhea regardless of oral or intestinal route, and now we do not recommend LECIG if a patient has a history of diarrhea with oral entacapone.

The tendency of levodopa accumulation during the day, which is a well-known phenomenon with oral entacapone [23], could also be a problem with LECIG infusion, according to its pharmacokinetic profile in comparison with that of LCIG, when both morning doses and continuous flow rates were decreased [14]. However, in a simulation of levodopa concentrations, unchanged morning dose and 35% decrease of continuous flow rate, compared with LCIG doses, provided stable levodopa levels without accumulation [24]. In the present study, morning doses were unaltered in the switch group, but flow rates were not decreased as much as 35%. A previous analysis of three datasets comprising 98 patients on LCIG revealed that about half of the patients had worse motor performance in the afternoon/evening than the first hours of the waking day [25]. Thus, levodopa accumulation may be beneficial for some patients, while others will likely become more dyskinetic in the evenings. In the case of evening dyskinesias, up to three different flow rates may be used in the LECIG pump, so the afternoon dosage may be adjusted to a lower rate. Two of our 24 patients used a slightly lower infusion rate in the afternoons.

Many patients on LCIG infusion use an oral dose of sustained-release levodopa at bedtime to try to keep some nighttime dopamine level, at least for a few hours [26]. With LECIG, it may not be necessary to keep the oral bedtime dose because the decline of levodopa concentration after disconnecting the pump and flushing the tube in the evening is slower for LECIG than LCIG [14]. This may be beneficial for patients during sleep but may also cause dopaminergic side-effects at nighttime. These aspects, as well as 24 h LECIG infusion, deserve more studies.

Most of the patients in the present study used supplementation with vitamin B12 and folic acid already when initiating LECIG. This makes homocysteine analyses redundant but should be prospectively studied in the future. Some increase in the number of patients using antipsychotics after initiation of LECIG was noted (from 8.3% to 16.7%). Hallucinations are common in advanced PD but may be elicited by dopaminergic drugs. Further prospective studies are needed to investigate any increased frequency of hallucinations during LECIG therapy.

There are certain limitations with the present observational study: it is small, open-label, noncontrolled, short-term, and includes a somewhat selected patient sample. Patients who opted to initiate LECIG were highly motivated and eager to try “the new pump”. The results cannot be generalized to the most severely disabled or demented PD patients.

Nevertheless, we think that our results are generalizable to the population with a clinical indication for LCIG/LECIG, except for those with previously intolerable side-effects from entacapone. According to the previous study of LECIG in 11 patients, COMT genotype did not seem to alter bioavailability [19].

The development of LECIG is a logical step in optimizing levodopa in advanced PD, as commented already in 2012 in an Editorial by Fabian Klostermann [27]. Future studies will analyze efficacy and safety more extensively and cost–benefit analyses, especially in comparison with LCIG. The cost of one syringe of LECIG is lower than one cassette of LCIG in Sweden today, but a syringe only contains 47 mL, corresponding to 1250 mg of levodopa equivalents, using 33% as a conversion factor, compared to the LCIG cassette containing 100 mL, i.e., 2000 mg of levodopa. At present, nationwide registry studies on LECIG are planned in several EU countries to gain more knowledge on this new treatment.

To conclude, the new LECIG formulation has been shown to be possible to use for at least a year in patients with PD. Patients are generally happy with the size of the pump, but technical improvements of the software are warranted, as well as larger, prospective studies.

## Figures and Tables

**Figure 1 jpm-11-00254-f001:**
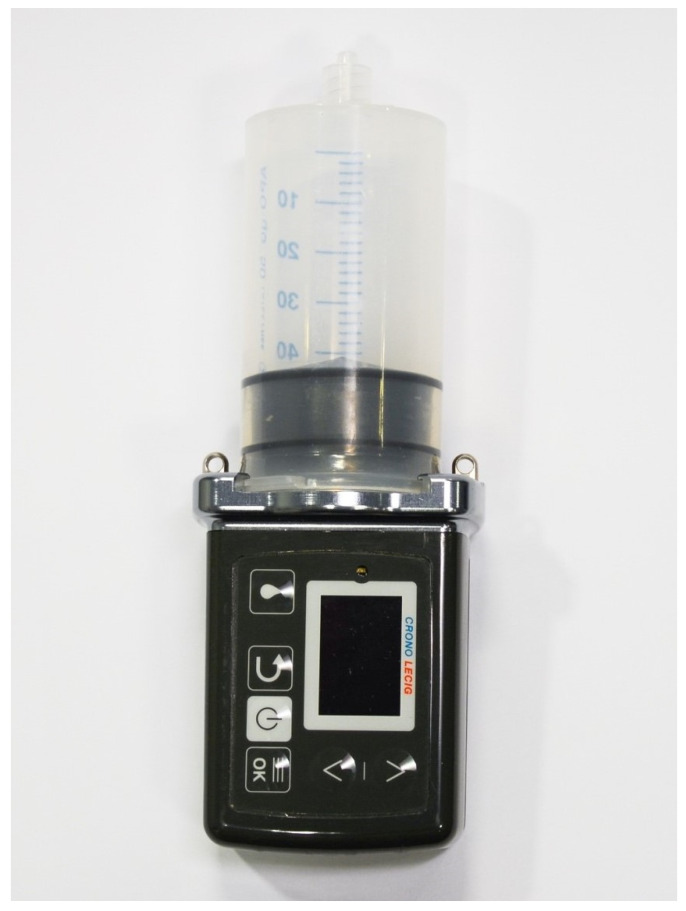
The levodopa–entacapone–carbidopa intestinal gel (LECIG) pump.

**Figure 2 jpm-11-00254-f002:**
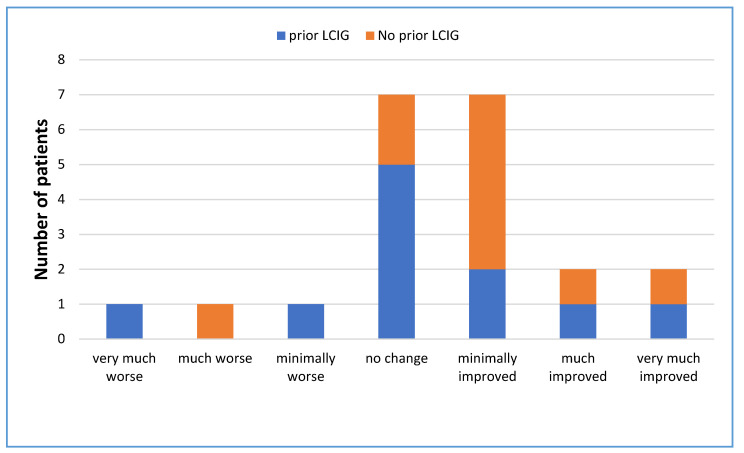
Patient-perceived efficacy of LECIG on PD symptoms (*n* = 21).

**Figure 3 jpm-11-00254-f003:**
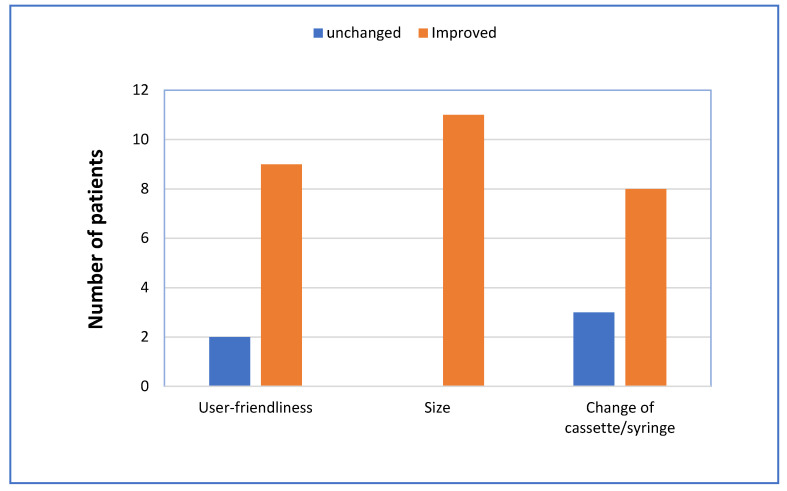
Comparison of pump characteristics by patients who switched from LCIG to LECIG (*n* = 11).

**Table 1 jpm-11-00254-t001:** Demographics based on data collected from medical records (*n* = 24).

Variable	Statistics
Age in years, median (range)	71.5 (45–78)
Gender (male:female)	13:11
PD duration in years, median (range)	15.5 (6–27)

**Table 2 jpm-11-00254-t002:** Characteristics of LECIG infusion therapy (*n* = 24).

Characteristic	Number (Range)
PEG-J tube: T-port:NJ tube, *n*	21:2:1
Device complications, *n*	5
Titration as inpatient: outpatient: video, *n*	17:6:1
Multiple flows daytime: daytime and at night, *n*	2:4
Adverse effects, *n*	7
Discontinuation due adverse effects, *n*	4
Deaths during treatment, *n*	2
LECIG duration in days, median (range).	305 (14–578)

NJ, nasojejunal.

**Table 3 jpm-11-00254-t003:** Treatments with other dopaminergic Parkinson’s disease (PD) medications prior to and during LECIG therapy.

PD Medication	Number of Patients, Prior to LECIG	Number of Patients during LECIG
Dopamine agonist (oral/transdermal: subcutaneous infusion)	13 (11:2)	9 (9:0)
Oral sustained release levodopa-DDCI	17	19
Amantadine	3	2
Levodopa-carbidopa-entacapone	6	0
LCIG	12	0
MAO-B inhibitor	5	0
Oral Levodopa-DDCI	10	0
LC-5	3	0

DDCI, dopa decarboxylase inhibitor; LC-5, levodopa–carbidopa microtablets; LCIG, levodopa-carbidopa intestinal gel; MAO-B inhibitor, monoamine oxidase B inhibitor.

**Table 4 jpm-11-00254-t004:** LECIG infusion rates (*n* = 24).

	First	Latest
MD (mL)	7 (2–12.5)	7 (2–12.5)
CD (mL/h)	2.5 (0.9–5)	2.7 (1.05–5)
CN (mL/h)	2.2 (1–3.1)	2.8 (2–3.2)

MD, morning dose; CD, continuous daytime infusion rate; CN, continuous nighttime infusion rate.

**Table 5 jpm-11-00254-t005:** Comparison between levodopa–carbidopa intestinal gel (LCIG) and LECIG (first and latest) dosage (*n* = 12).

	LCIG	LECIG, First	LECIG, Latest
MD (mL)	6 (2.5–11)	6 (2.5–12)	5.5 (2.5–12)
CD (mL/h)	3.85 (1.5–6.7)	2.5 (1–5)	2.6 (1.4–5)

MD, morning dose; CD, continuous daytime infusion rate.

**Table 6 jpm-11-00254-t006:** Non-dopaminergic PD treatment prior to and during LECIG (*n* = 24).

Non-Dopaminergic PD Medications	Prior: *n* (%)	During: *n* (%)
Vitamin B12	19 (79%)	20 (83%)
Folic acid	16 (67%)	17 (71%)
Antidepressants	15 (63%)	14 (58%)
Laxatives	13 (54%)	14 (58%)
Midodrine	3 (13%)	3 (13%)
Etilefrine	2 (8%)	1 (4%)
Fludrocortisone	2 (8%)	2 (8%)
Donepezil	1 (4%)	1 (4%)
Quetiapine	2 (8%)	4 (17%)

**Table 7 jpm-11-00254-t007:** Patient-perceived ability to perform daily activities and quality of life after initiation of LECIG (*n* = 21).

	Improved (*n*)	Unchanged (*n*)	Worsened (*n*)	I Do Not Know (*n*)
Ability to perform daily activities	12	6	3	
Quality of life	13	5	2	1

## Data Availability

The data presented in this study are available on request from the corresponding author. The data are not publicly available due to privacy.

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
