# Peer review of "Initial Experience of the Levodopa–Entacapone–Carbidopa Intestinal Gel in Clinical Practice"

_jpm, 2021, doi:10.3390/jpm11040254_

Round 1
Reviewer 1 Report
At present, I have not concerns over the content of this article. The study presented by the authors have been well documented. The study seems a bit preliminary and has lot of limitations (as accepted and discussed by the authors) specially the fact that the study is based only on data from 24 patients in last few months. However, I do not see any major concerns.
Author Response
We thank the reviewer for these comments. We have now elaborated on plans for larger studies on page 9, line 262.
Reviewer 2 Report
It is a good observational study, larger prospective studies are needed.
Treatment of the patients with advanced Parkinson’s Diseases with motor fluctuations is a challenge for neurologists. Levodopa-carbidopa intestinal gel (LCIG) is a good option known for more than 20 years.
Inhibitors of catechol-O-methyl- 40 transferase (COMT) as oral therapy are well-known adjunctive to levodopa, so the logical step in LCIG development was the ad of entacapone to Levodopa-carbidopa intestinal gel - the levodopa-entacapone-carbidopa intestinal gel (LECIG).
The LECIG have the advantage of decreasing the levodopa necessary by 33% and a smaller and more wearable device, more agreeable to the patients.
Also, this patients should not use anymore vitamin B supplements in order to prevent polyneuropathy, because COMT inhibitors also prevent the methionine cycle from using B vitamins for formation of homocysteine.
The side effects of entacapone like diarrhoea were observed, in spite of the new mode of administration as intestinal gel also the increased frequency with hallucinations.
Author Response

(The authors gave the same response as above.)

Reviewer 3 Report
As pointed out by the authors themselves, the sample size used in this study is quite small, the study is also quite short with biased sample population. I appreciate that the authors are cognizant about these facets of their study and clearly a larger blinded and controlled study is warranted. The data presented is preliminary and not sure if statistically significant. Other than user friendliness of the pump used for LECIG dosage, there isn't a significant advantage of using LECIG over LCIG as seen in this study. Here are some points that the authors should address -
- Authors should explain/elaborate why they chose entacapone over tolcapone
- They should present statistical analysis of the data
- Is there is plan for a larger controlled study using LECIG and authors should consider elaborating on some of the details
- Authors present a reason for some of the side effects like diarrhea, however, they don't explain the reasons for other side effects like hallucination
